# Storing and managing water for the environment is more efficient than mimicking natural flows

**Sarah E. Null** [1,2] ✉, **Harrison Zeff**[3], **Jeffrey Mount**[2], **Brian Gray**[2], **Anna M. Sturrock** [4], **Gokce Sencan**[2], **Kristen Dybala** [5] **& Barton Thompson**[6]

Dams and reservoirs are often needed to provide environmental water and maintain suitable water temperatures for downstream ecosystems. Here, we evaluate if water allocated to the environment, with storage to manage it, might allow environmental water to more reliably meet ecosystem objectives than a proportion of natural flow. We use a priority-based water balance operations model and a reservoir temperature model to evaluate 1) pass-through of a portion of reservoir inflow versus 2) allocating a portion of storage capacity and inflow for downstream flow and stream temperature objectives. We compare trade-offs to other senior and junior priority water demands. In many months, pass-through flows exceed the volumes needed to meet environmental demands. Storage provides the ability to manage release timing to use water efficiently for environmental benefit, with a co-benefit of increasing reservoir storage to protect cold-water at depth in the reservoir.

Dams and reservoirs degrade freshwater ecosystems by blocking access to high-quality upstream habitat and altering hydrology, geomorphology, and biogeochemistry downstream of dams[1,2]. To counteract these changes, environmental flows are sometimes released from reservoirs to augment flow, maintain water quality, and sustain aquatic species and habitats[3–5]. This creates a paradox where the major contributor to the decline of freshwater ecosystems—dams and their reservoirs—also holds the key to their survival. This paradox begs an important question: can reservoir storage be allocated and managed explicitly to revive river health?

Environmental flow prescriptions have focused on the *effectiveness* of environmental water—the degree to which flows produce desired results[6–10]. When implemented, water is typically withheld from appropriation to farms and cities to comply with water quality, flow, and endangered species regulatory requirements or negotiated compromises. This makes flow requirements a constraint on water operations, rather than a priority objective in multipurpose water management[11]. The *efficiency* of environmental water—the ability to

accomplish ecosystem objectives with the least water, time, money, and effort—has often been overlooked. Some combination of water allocated to the environment, with storage to manage it—which we call an *environmental water budget*— might allow environmental water to be used more efficiently. This would make environmental water an operational priority with human water uses in large, multi-purpose reservoirs[12].

Few dams have been built specifically for environmental water storage in the USA. An example is Nevada's Marble Bluff Dam on the Truckee River, which provides water for endangered fish migration via the Pyramid Lake fishway and curtails streambed erosion caused by Pyramid Lake level decline[13]. More precedent exists for allocating storage space for environmental water in Australia. The 2007 Commonwealth Water Act allows water to be purchased and stored for environmental water entitlements[14]. Environmental water can be released to augment streamflow, stored as carryover for the following year, or traded for equal or greater environmental benefit in regulated basins. Carryover increases the likelihood of environmental water

[1]Department of Watershed Sciences, Utah State University, Logan, UT, USA. [2]Water Policy Center, Public Policy Institute of California, San Francisco, CA, USA. [3]Department of Environmental Sciences and Engineering, University of North Carolina, Chapel Hill, NC, USA. [4]School of Life Sciences, University of Essex, Colchester, Essex, UK. [5]Point Blue Conservation Science, Petaluma, CA, USA. [6]Stanford Law School & Doerr School of Sustainability, Stanford University, Palo Alto, CA, USA. ✉e-mail: sarah.null@usu.edu

availability in dry years and allows infrequent, high-magnitude pulse flows to reintroduce hydrologic variability[15]. Similar ideas have been proposed in parts of the USA. For example, California's 2014 Water Storage Investment Program provides $2.7 billion to support new surface and underground water storage for public benefits, including storage and management of environmental water[16]. Sites Reservoir, a proposed off-stream surface storage project, is under consideration and, if built, would provide 296 million cubic meters (Mm³) of storage, with around 17% of inflows passed through (or exchanged) to meet downstream environmental water demands. Additional proposals include establishing groundwater banks or raising existing dams to create storage space for environmental water[16].

We evaluate environmental water efficiency and trade-offs to other water demands from two management approaches: (1) pass-through of 10–40% of inflows through reservoirs for downscaled natural flows[17,18], or (2) allocating 10–40% of inflow and 10–40% of reservoir storage capacity for environmental demands[18] (not including dead pool and seasonal flood storage space). In some runs, we constrain minimum reservoir storage to increase the likelihood of cold water in storage for downstream water temperature objectives. These alternatives are exemplified by new flow objectives for California's San Joaquin River. The California State Water Resources Control Board adopted amendments that require water users to pass through or release an average of 40% of February–June unimpaired flows on the Lower San Joaquin River and its tributaries if water users fail to negotiate 'voluntary agreements' to reduce water use that is approved

by the Board[17]. Further upstream, the San Joaquin River Restoration Program has a Restoration Administrator who manages a percentage of unimpaired inflow, which can be stored and released to provide ecosystem benefits[18].

While flexible environmental water is broadly beneficial for improving ecosystem function and important across multiple taxa, we focus on the four runs of Chinook Salmon (*Oncorhynchus tshawytscha*) that spawn in and/or out-migrate through the mainstem Sacramento River to demonstrate the concept and elucidate potential benefits and trade-offs. One of these species, the endangered Sacramento River winter-run Chinook Salmon, is at high risk of extinction and in need of urgent protection[19,20]. We synthesize flow and temperature requirements with three environmental water demand objectives, ranked as follows: (1) environmental baseflows to account for existing minimum instream flows and water quality standards[21,22], (2) flow shaping, where water for a fall pulse, winter pulse, and spring recession (Supplementary Table 1) can be shaped by water managers to mimic aspects of flow regimes that support ecological function[6,7,23,24], and (3) *optimal* water temperatures, which require colder water than merely *suitable* temperatures and are deemed to be more protective and more likely to promote salmonid recovery.

We developed a simple, priority-based water balance operations model coupled with a one-dimensional reservoir temperature model that stratifies vertically, based loosely on California's 5.55 billion cubic meters (Bm³) (4.5 million acre-feet) multipurpose Shasta Reservoir (Fig. 1). We ran the model on a monthly timestep for water years

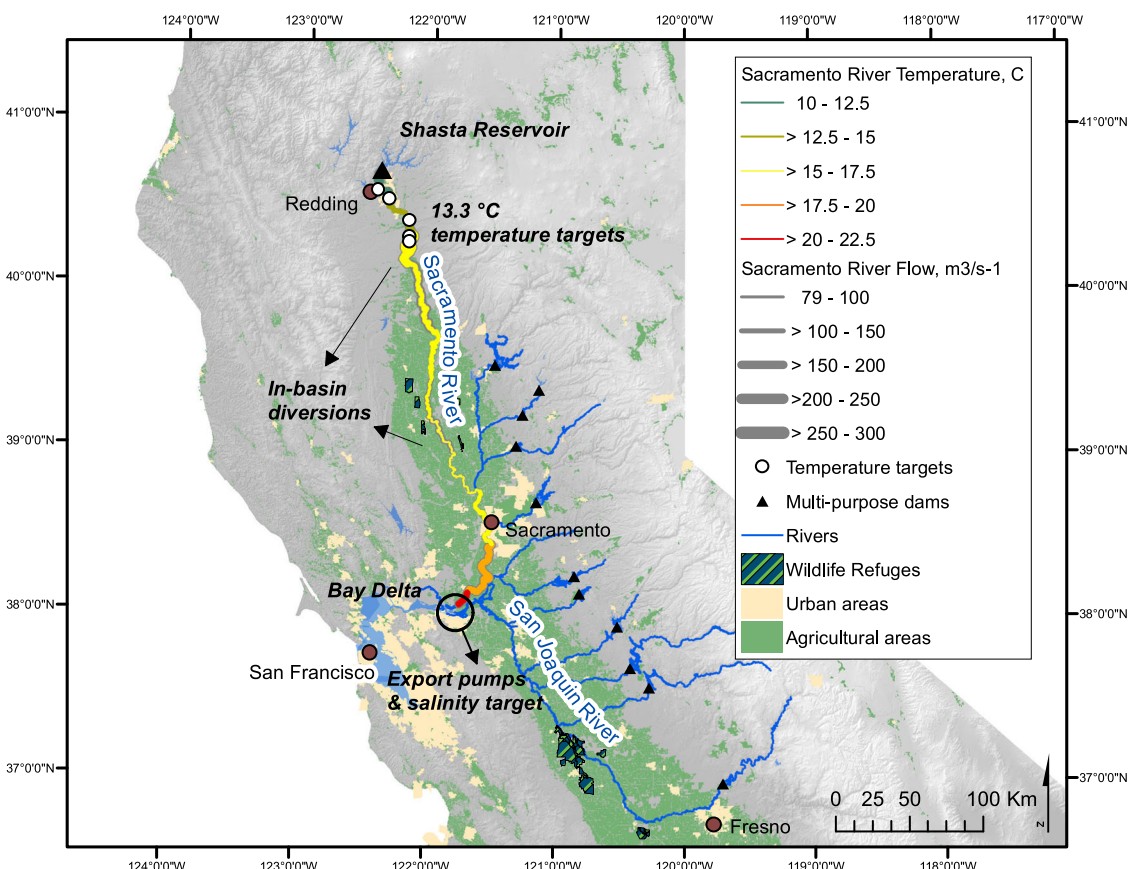

**Fig. 1 | Sacramento River flows and temperatures on July 1, 2021.** Sacramento streamflows and water temperatures are from the following California Data Exchange Center (CDEC) monitoring stations: Sacramento River at Shasta Dam-USBR (SHA), Shasta Dam-Water quality (SHD), Keswick (KWK), above Clear Creek (CCR), Balls Ferry Bridge (BSF), Jellys Ferry (JLF), Bend Bridge (BND), Red Bluff Diversion Dam (RDB), Hamilton City-main channel (HMC), Ord Ferry-main channel

(ORD), Butte City (BTC), Colusa (COL), Wilkins Slough (WLK), Verona (VON), Freeport (FPT), Hood (SRH), Rio Vista Bridge (RVB), and Emmaton-USBR (EMM). Figure data sources: CDEC, USGS National Hydrography Dataset, California Dept of Fish and Wildlife, California Dept. of Transportation, California Dept of Water Resources, Consortium of International Agricultural Research Centers (CGIAR), and Sacramento River Temperature Task Force.

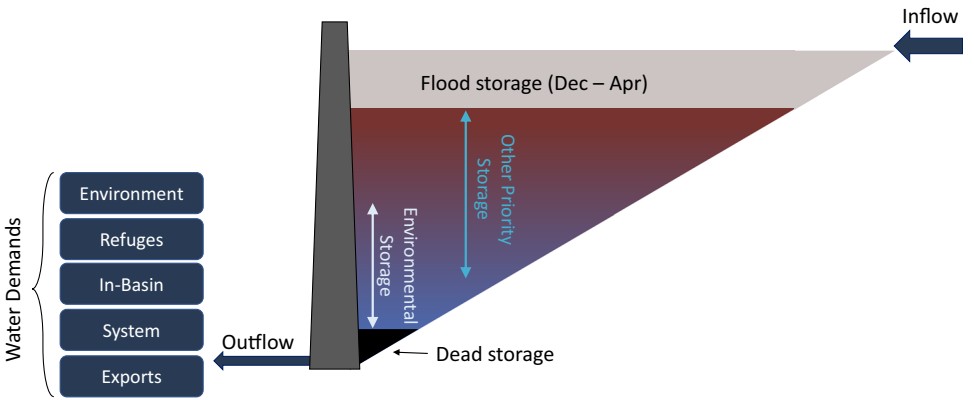

**Fig. 2 | Modeled inflow and environmental water storage with stylized water demands.** This diagram depicts different allocations of reservoir storage capacity for the environment and other water demands. Reservoir colors represent summer reservoir temperature stratification, with warmer water at the surface and cooler water at the depth.

1996–2021 to capture the range of historical hydrologic variability. We estimate storage and releases for (1) environmental objectives, (2) wildlife refuge water demands—which have water rights and so are separate from environmental demands, (3) in-basin urban and agricultural uses, (4) system water for salinity maintenance through the Sacramento-San Joaquin Delta, and (5) out-of-basin exports (Fig. 2). The first four demands—environmental, refuge, in-basin urban and agricultural, and system water for salinity maintenance—share senior water right priority in the model. The last—out-of-basin exports—is junior to the other demands[21,22]. Water may be carried over if storage capacity exists, although carryover water is first to spill during wet periods, then spill occurs in reverse-priority order. Our approach applies to large, multi-purpose reservoirs, with highly variable seasonal and interannual inflows, temperature stratification during summer, minimum operational levels (dead pool), and seasonal flood storage requirements[25].

## Results

### Inflow pass-through performance

Performance on environmental baseflow and flow shaping objectives generally improve as a larger portion of inflow passes through the reservoir (Fig. 3A, B). Larger pass-through flows come closer to mimicking natural flows and variability. Pass-through of 10% of inflows through the reservoir fails to meet environmental demands. In dry years, environmental baseflow deliveries average 44% of demand and flow shaping deliveries average 20% of demand (Fig. 3A). In wet years, the 10% pass-through delivers 32% of environmental baseflow demand, on average, and flow shaping deliveries average 30% of demand (Fig. 3B). Environmental baseflow demands are met more often in dry years than wet years because the regulatory requirements that environmental baseflows represent are smaller in dry years than wet years[21,22,26]. For all years with 10% pass-through, the interquartile range of environmental baseflow shortages is 203 – 780 Mm³/yr (165-632 thousand acre-feet per year [taf/yr]) (Fig. 4A), and flow shaping shortages range from 617–762 Mm³/yr (500–618 taf/yr) (Fig. 4B).

With 40% pass-through, flow performance improves. In that alternative, environmental baseflow deliveries average 90% of demand and flow shaping deliveries average 76% of demand in dry years (Fig. 3A), and environmental baseflows average 94% of demand and flow shaping average 68% of demand in wet years (Fig. 3B). The interquartile range of shortages is 12–208 Mm³/yr for baseflows (Fig. 4A) and 153–339 Mm³/yr for flow shaping (Fig. 4B). However, in many months, pass-through flows exceed the volumes needed to meet environmental baseflow and flow shaping demands. Without storage, there is no ability to manage the timing of releases to use water efficiently for environmental benefit.

The pass-through approach results in clear trade-offs among environmental objectives. As the portion of pass-through increases, performance on temperature objectives worsens as reservoir storage drops and the cold-water pool is depleted (Fig. 3A–D, Fig. 4). In dry years, optimal stream temperature objectives are attained about 68% of all months with 10% pass-through flows, but only about 53% of the time when pass-through allocations are increased to 40% (Fig. 3A). In wet years, the trade-off between flow and temperature objectives is diminished, but not eliminated. In those years, stream temperature objectives have attained an average of 73% of months with 10% pass-through, and an average of 68% of months with 40% pass-through (Fig. 3B). Pass-through flows in conjunction with constraining minimum reservoir storage is marginally useful to preserve cold-water at depth in the reservoir (Fig. 3C and D). Since environmental deliveries are a percentage of inflows, environmental baseflow, and flow shaping objectives do not change when minimum reservoir storage is constrained.

Larger portions of environmental pass-through worsen shortages for in-basin urban and agricultural, refuge, system water, and out-of-basin export demands (henceforth called 'other water demands') (Fig. 5A-D). The senior water priorities (in-basin urban and agricultural, system water, and refuges) average over 90% of demands, even with 40% inflow pass-through. In dry years, system water and in-basin demands average 88% of demands, and refuge demands average over 99% of demands (Fig. 5A). But in critically dry years, even in-basin and system water demands experience large average annual shortages (Supplementary Figure 1) caused by higher in-basin demand. In contrast, environmental shortages are largest in wet year types because environmental baseflows—which are set by regulatory criteria—are the largest. Average dry year shortages to junior export demands are acute, deliveries fall to less than 30% of demands (Fig. 5A).

When minimum reservoir storage is constrained to 1.54 Bm³ (1.25 maf) to preserve cold water deep in the reservoir, average deliveries fall by 6–9% for senior water demands, depending on the portion of inflow allocated for pass-through (Fig. 5C, D). Average deliveries remain over 85% for all other demands, even when 40% of flows are passed through the reservoir. Junior priority export demands experience considerable shortages with minimum reservoir storage when pass-through is 40%, with deliveries averaging close to 60% for all year types and declining to 27% during dry years (Fig. 5C).

### Environmental water budget performance: inflow plus storage capacity

Allocating a portion of reservoir inflow with storage capacity is invaluable for using environmental water efficiently because water can be stored seasonally or interannually to target environmental

## Pass-through

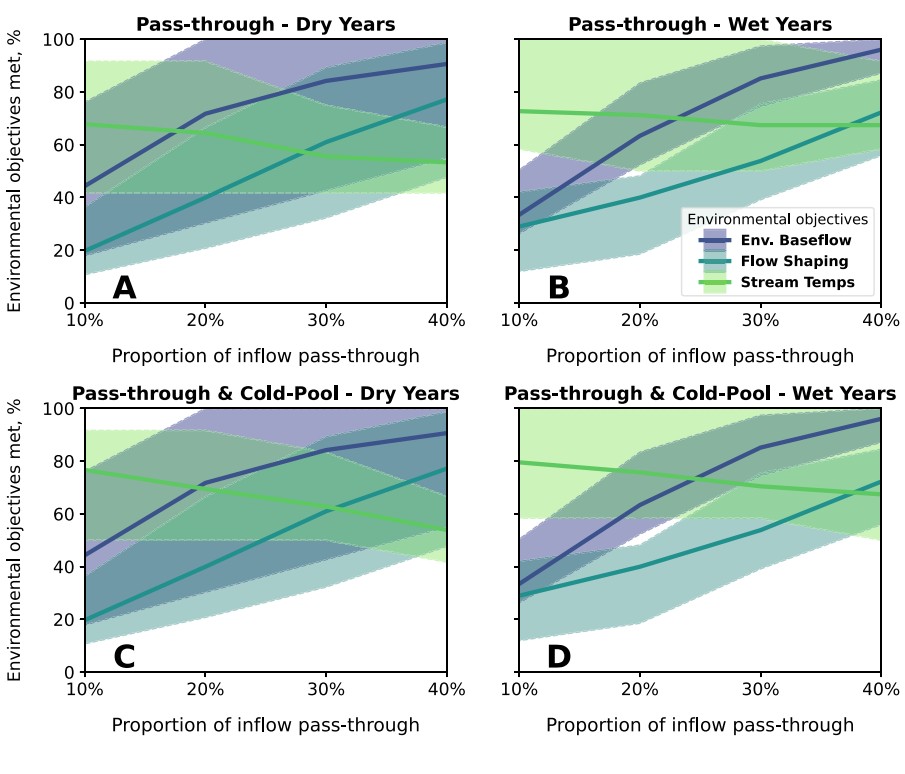

## EWB Storage

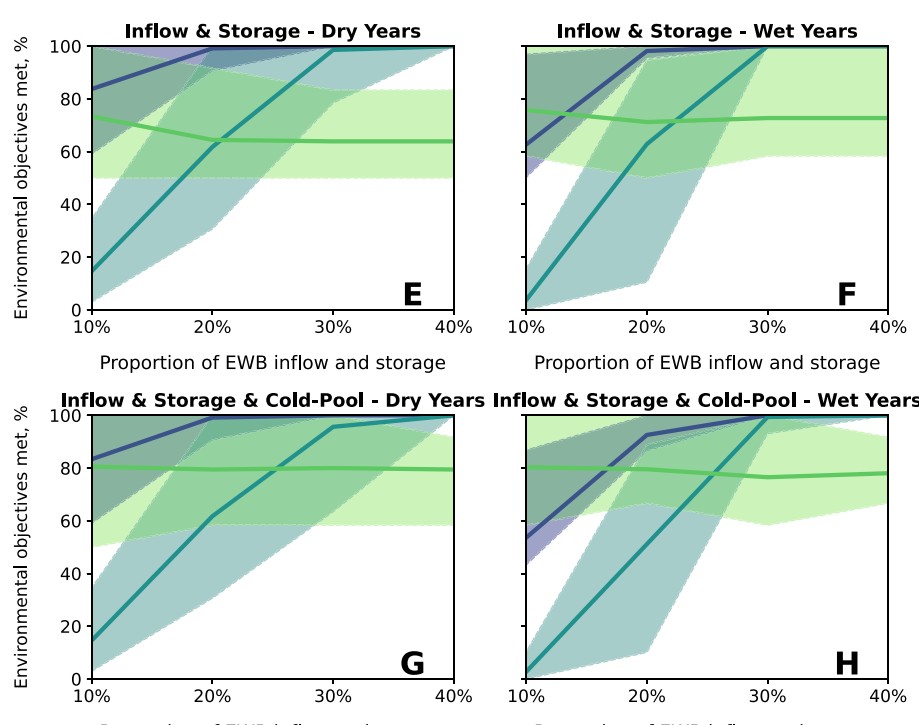

**Fig. 3 | Average percentage of months that environmental water demands are met (lines) and range of months that environmental objectives are met (shaded area) from 1996–2021.** Panels show environmental water management alternatives, with pass-through flows in dry years (**A**) and wet years (**B**), passthrough flows with 1.54 B m³ of cold-water storage in dry years (**C**) and wet years (**D**), and an Environmental Water Budget (EWB) that includes equal portions of inflow and reservoir storage for dry years (**E**) and wet years (**F**), and EWB with 1.54 Bm³ of cold-water storage in dry years (**G**) and wet years (**H**). Dry years include critically dry, dry, and below-normal Sacramento River Index water year types, and wet years include above-normal and wet year types.

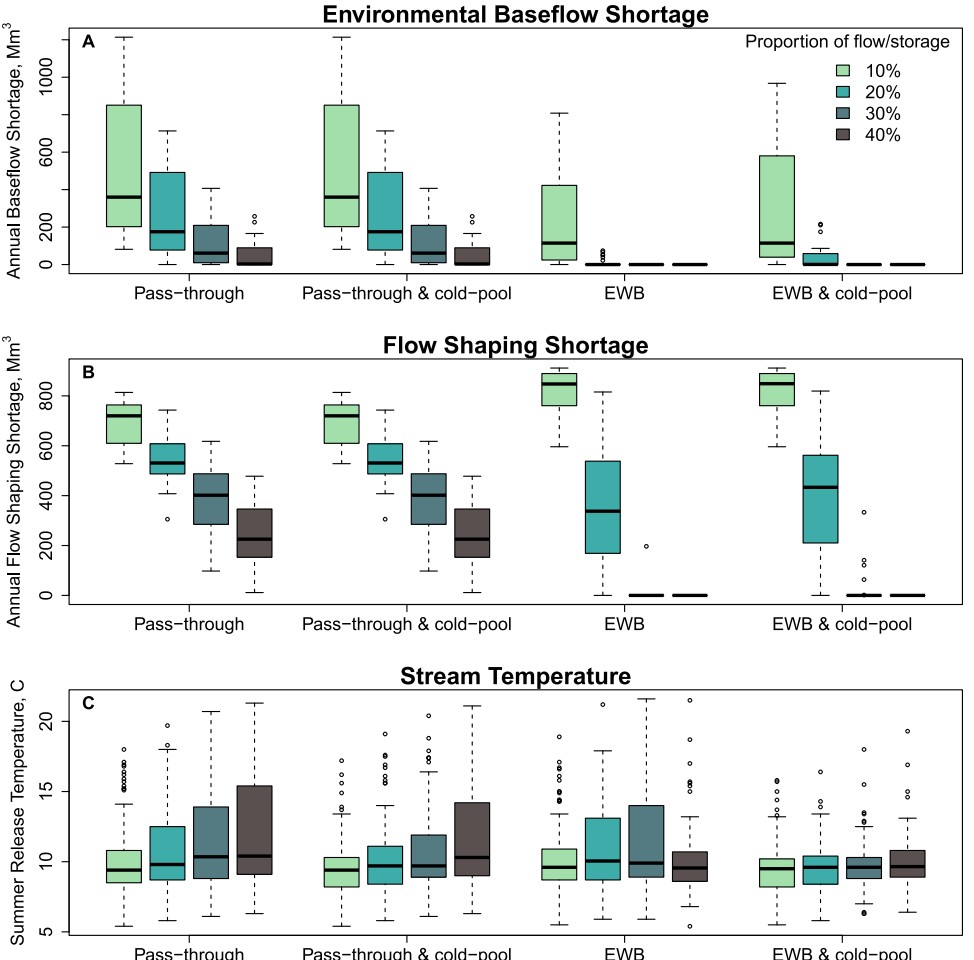

**Fig. 4 | Environmental objective performance by water management alternative and proportion of assets.** Panels show environmental baseflow shortages (**A**), flow shaping shortages (**B**), and July – September reservoir release temperatures (**C**) for all environmental water management alternatives. EWB = Environmental Water Budget. Boxes show upper and lower quartiles, bold horizontal lines show medians, whiskers show 1.5 times the interquartile range, and dots show outliers.

demands. For alternatives with 10% of inflow and 10% of reservoir capacity allocated for environmental management, reservoir inflows are insufficient to meet all flow demands (Fig. 3E, F). Environmental baseflows are almost always delivered, but there is not enough water for flow shaping demands and little buffer for critically dry periods. The interquartile shortage range is 28–375 Mm³/yr (23–304 taf/yr) for environmental baseflows and 763–888 Mm³/yr (618–720 taf/yr) for flow shaping demands (Fig. 4A, B). With 30% allocation of inflows and 30% of storage for the environment, 99% of environmental baseflows and 96% of flow shaping demands are delivered, on average, for wet and dry years.

Storage for environmental water enables temperature objectives to be met more frequently—both with and without minimum storage constraints to protect the cold-water pool. Storing environmental water increases reservoir storage. Average stream temperature objectives are met 64–73% of months in dry years (Fig. 3E) and 71–76% of months in wet years (Fig. 3F) for all modeled proportions of environmental water and storage. A minimum reservoir storage constraint further improves stream temperature objectives, since minimum reservoir storage for cold-water preservation is effectively a third asset for environmental management. Environmental storage capacity, dedicated inflow, and a minimum storage requirement maintain optimal stream temperatures for about 77–80% of months across all alternatives and water year types (Fig. 3G, H). With minimum storage to increase the likelihood of cold water at depth in the reservoir,

summer stream temperatures are consistent among 10% to 40% flow and storage allocations, with summer median reservoir release temperatures ranging from 9.5–9.7 °C, and an interquartile range of 8.2 to 10.7 °C (Fig. 4C).

When 30% or more of inflow and storage capacity is allocated to the environment in dry years, junior water uses face severe cutbacks (Fig. 5E–H). When 40% of inflow and 40% of reservoir storage capacity are allocated to the environment in dry years, average deliveries near 80% for system water and in-basin urban and agricultural uses, and average more than 95% for refuges (Fig. 5E). Increasing minimum reservoir storage to manage the cold-water pool has a large effect on other water demands because constraining minimum reservoir storage effectively shrinks storage capacity for these demands and reduces the total volume of water that can be carried over from wet years for use in later years (Fig. 5G, H).

## Example of environmental water storage performance in 2019–21

The three-year drought sequence beginning in 2019—a wet year followed by dry and critically dry years in 2020 and 2021—illustrates the benefits of dedicating inflow and a portion of storage capacity to environmental demands (Fig. 6). In this example, we compare allocating 30% pass-through (Fig. 6, left side) with 30% of inflow and 30% of storage space for the environment (Fig. 6, right side). Both alternatives include a minimum storage requirement of 1.54 Bm³ to increase

## Pass-through

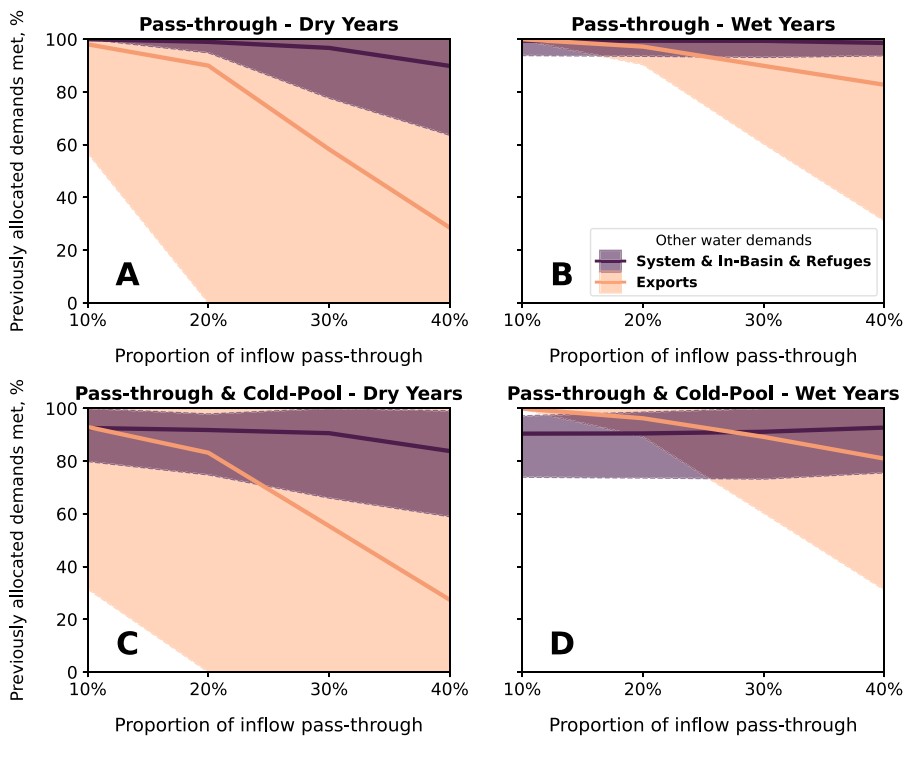

## EWB Storage

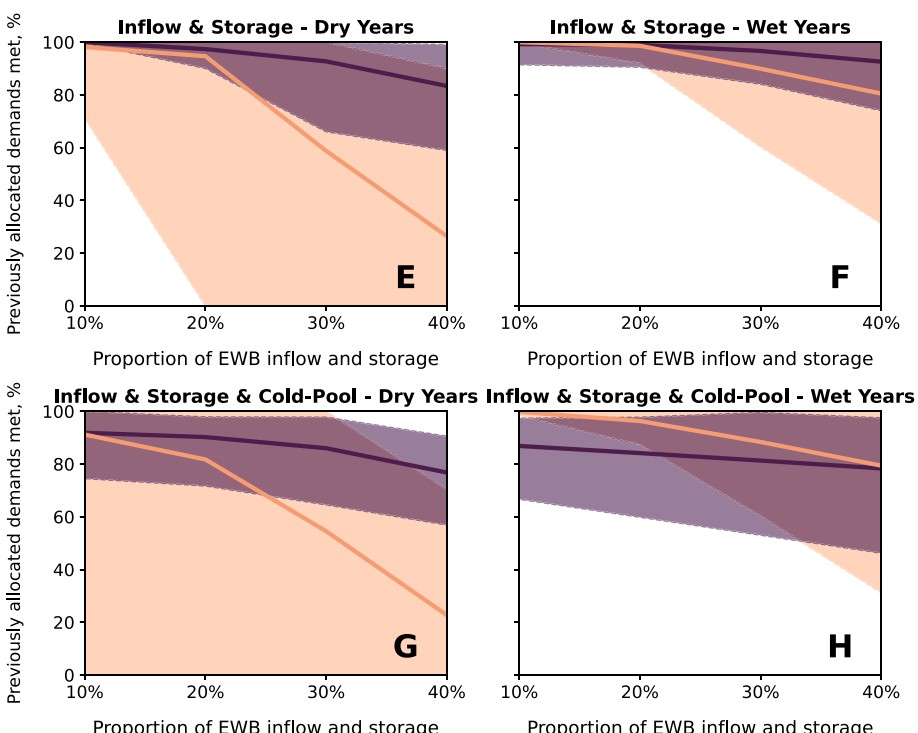

**Fig. 5 | Average percentage of months that in-basin agricultural and urban, wildlife refuge, system water, and out-of-basin export water demands are met (lines) and range of months that each demand is met (shaded area) from 1996–2021.** Panels show environmental water management alternatives, with pass-through flows in dry years (**A**) and wet years (**B**), passthrough flows with 1.54 Bm3 of cold-water storage in dry years (**C**) and wet years (**D**), and with an Environmental

Water Budget (EWB) that includes an equal portion of inflow and reservoir storage for dry years (**E**) and wet years (**F**), and EWB with 1.54 Bm³ of cold-water storage in dry years (**G**) and wet years (**H**). Dry years include critically dry, dry, and below-normal Sacramento River Index water year types, and wet years include above-normal and wet year types.

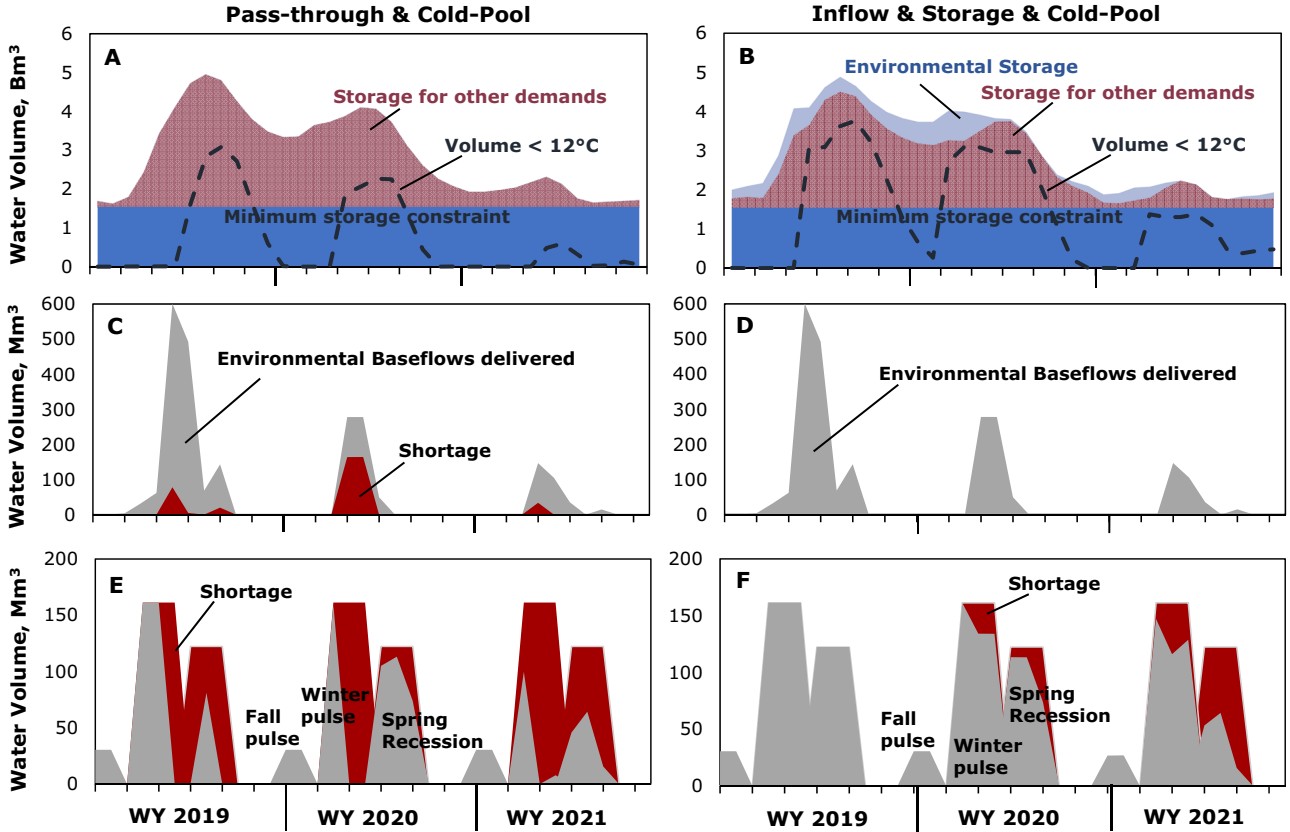

**Fig. 6 | Comparison between bypassing flows to mimic natural flows and storing water for the environment.** Environmental storage and deliveries with 30% pass-through and 1.54 billion m3 minimum reservoir storage for cold-water management (left side), and 30% of inflow, 30% of storage capacity, and 1.54 billion m3 minimum reservoir storage for cold-water management (right side) during the 2019–21 water year (WY) drought. Panels **A** and **B** show water storage and volume of water <12 °C, panels **C** and **D** show environmental baseflow deliveries and shortages, and panels **E** and **F** show flow-shaping deliveries and shortages.

the likelihood of cold water in the reservoir. Reservoir storage, the volume of water less than 12 °C, and water deliveries for environmental baseflows and flow shaping illustrate differences between the approaches.

Environmental demands are lowest during the summer dry period, so storage capacity allows water to be stored throughout summer when other water demands draw down reservoir storage (Fig. 6, right side). Increased summer storage provides dual ecosystem benefits, improving the chance of meeting winter peak and fall pulse flow shaping objectives while also raising reservoir storage to meet late summer and fall temperature objectives. With minimum reservoir storage constrained at 1.54 Bm³, the cold-water pool can be managed to meet downstream temperature standards until late summer to early autumn (Fig. 6B). Carryover storage throughout 2019 and again in the winters of 2020 and 2021 is sufficient to meet environmental baseflow demands fully (these demands are lower in dry years than wet years) (Fig. 6D). As reservoir inflows are diminished with prolonged drought from 2019 through 2021, the 30% inflow allocation cannot meet flow shaping demands, with shortages of 13% (121 Mm³) in 2020 and 37% (333 Mm³) in 2021 (Fig. 6F). The significant shortages to other demands are shown in Figs. 5G and 5H and Supplemental Fig. 2. In 2021, these shortages range from 12% for wildlife refuges to 92% for exports. Overall, non-environmental demands have an average water shortage of 45% of their annual demands during 2021, greater than would have occurred from only an environmental pass-through scenario (Fig. 5C, D).

**Trade-offs between environmental demands and other water demands.** Trade-offs between environmental and other water

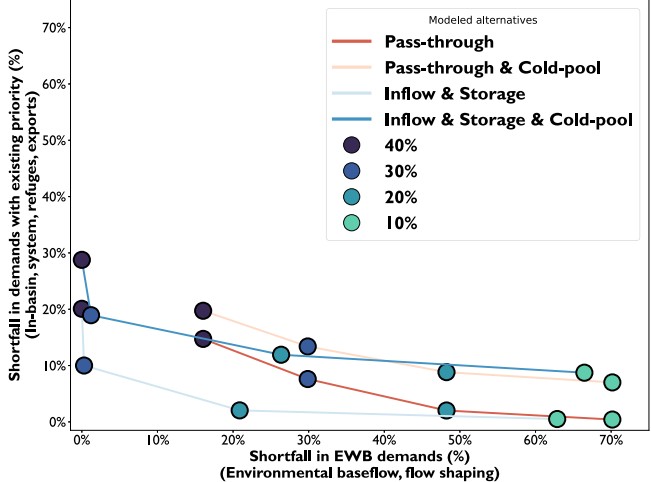

**Fig. 7 | Trade-offs between environmental water shortage and other demand shortage across 26 years of hydrologic conditions.** Environmental deliveries were modeled as a portion of inflow for pass-through or a portion of inflow with reservoir storage capacity. Percentages shown with the dots are the share of the inflow or equal shares of inflow and storage capacity.

demands highlight the benefits of an environmental water budget, versus reservoir pass-through (Fig. 7). When the environment is allocated 10% of inflows—whether as pass-through or with storage capacity—most shortages accrue to environmental objectives. Environmental

baseflows—required to meet regulatory flow and water quality standards—average about 8% of reservoir inflow for Shasta Reservoir[27]. Other water demands with water rights, which are prioritized in water management, receive the remaining average of 92% of reservoir inflows. As environmental allocations increase, shortages of other water demands increase. However, allocating inflows with storage capacity to manage water allows environmental water to be used more efficiently than pass-through. These alternatives shift tradeoff curves leftward in Fig. 7, toward a more optimal region with greater total benefit. For example, with 20% pass-through, 48% of flow objectives are unmet. The shortage drops to 21–26% when storage is used to manage environmental water, depending on the minimum reservoir storage for the cold pool. Storing water for the environment results in a much smaller chance of shortages for other water demands. These range from 2–9% with pass-through alternatives, and become 2–12% when the environment is allocated 20% of inflow and 20% of storage. We show that without storage to manage environmental allocations, larger pass-through flows are required to meet flow objectives.

When environmental storage and flow allocations exceed 30%, considerable shortages are incurred to other demands (Fig. 7). Allocations to the environment beyond this point have little environmental benefit and incur substantial shortages to other demands. This "knee", or breakpoint in trade-offs, suggests that 30% of inflow and storage for an environmental water budget is adequate in our simplified model to meet environmental baseflows and flow shaping objectives, while additional water and storage for the environment would improve the likelihood that water temperature targets are met. Breakpoints highlight promising areas for compromise, where decision-makers are more likely to cooperate[28], which merits further exploration with more detailed modeling.

## Discussion

Our study reveals important insights into how to operate a reservoir that sets environmental water demand as a primary objective, rather than as a constraint on water supply operations. Allocating a percentage of inflow and a percentage of operable storage space for environmental management is most efficient for meeting environmental and other objectives. In our system, allocating 30% of water for the environment is enough to meet baseflow and flow shaping objectives when reservoir capacity is allocated, but is insufficient without reservoir capacity (Fig. 4). Environmental storage capacity reduces trade-offs among environmental objectives (e.g., water temperatures versus environmental baseflow versus flow shaping) that occur from reservoir pass-through. Carryover storage could be used to provide higher flows in some years or to maintain cold water at depth in reservoirs, both of which benefit species survival[8,29–31]. However, dry year shortages are profound for junior priority export demands, exacerbating existing shortages from over-allocation of available water supplies[32]. Setting minimum reservoir levels improves water temperature management, albeit with trade-offs to other demands. While temperature management in reservoirs is always challenging, reservoir pass-through creates the greatest threat to reservoir cold-water pools. In fact, our modeling suggests that without environmental storage, allocating more water to the environment as reservoir pass-through results in a worse outcome for temperature. This approach should be avoided where temperature management is an objective. Our study demonstrates how to incorporate thermal regimes with environmental flows for more holistic environmental water management.

Our model is simple, intended as a proof-of-concept to understand and compare how portions of inflow and storage capacity allocated to the environment could benefit ecosystem objectives and impact other water demands. Our approach complements studies that prescribe environmental or functional flows[7,8,30]. It is not intended to be a guide for setting specific standards or determining the adequacy

of environmental flows to support species and ecosystem function. Sophisticated water management and water temperature models exist for California's water system[33–36] and most large river basins[37,38]. Those models could be applied to scrutinize and elucidate the potential benefits, tradeoffs, breakpoints, and impacts of inflows-plus-storage space allocation in real systems.

Storage capacity for carryover is instrumental in managing environmental water efficiently. Designer flows, which alter the timing of reservoir releases to benefit ecosystem objectives while maintaining the volume of water delivered to other water demands, implicitly use reservoir storage for environmental benefit[9,10]. In this way, designer flows have increased the flexibility of environmental water management, although they treat environmental water as a constraint on water supply and hydropower operations rather than an explicitly managed objective. An enlarged portfolio of environmental water management strategies like interannual carryover, water markets, in-lieu exchanges, and conjunctive management require environmental allocations and storage.

To successfully protect aquatic ecosystems, water assigned to environmental purposes must be an operational priority in large, multipurpose reservoir management, and have allocated assets[25]. Environmental water budgets could create this with a proportion of inflow, reservoir storage space to manage it, and sometimes minimum reservoir storage levels[39]. A designated trustee with the authority to allocate and release water, prioritize ecosystem objectives, and coordinate with all other relevant parties could administer these environmental assets. Water and funding to support environmental water budgets could come from incorporating water that is dedicated to environmental uses under existing regulations, negotiation of agreements to enhance these allocations, purchase, new storage infrastructure, water user fees, and government support[25].

While allocating assets to the environment for flexible management is a major change that could reduce water availability for other water users, it would also lessen regulatory uncertainty. Environmental water should be managed like a senior water right, with the release schedule integrated into reservoir operations. Under existing law, water users' obligations to comply with water quality standards, endangered species requirements, and other environmental laws take precedence over water supply for consumptive uses—i.e., this water generally carries top priority within each river system[11,40]. These regulations are managed with little margin for error, however, and environmental uses bear an inordinate risk of forecasting mistakes and operational errors. Moreover, for many water systems, environmental regulations are often relaxed during periods of acute shortage to make more water available for consumptive uses. All of these factors create uncertainty for the sustainable management of environmental water[39,41].

The above governance, policy, and funding mechanisms are not unprecedented. Examples in California include amendments that require water users to pass through or release an average of 40% of February–June unimpaired flows on the Lower San Joaquin River and its tributaries or voluntarily reduce water use[17]. A Restoration Administrator manages a percentage of unimpaired inflow farther upstream in the San Joaquin River, which can be stored and released to provide ecosystem benefits[18]. California's Water Storage Investment Program funds the environmental benefit portion of private water projects in exchange for environmental water storage and releases[16].

This study assumes storage space in an existing large reservoir for environmental management. Underground storage also provides opportunities for water trading and exchanges that would facilitate environmental water releases and carryover storage. Studies have shown that managed aquifer recharge in hydrologically connected groundwater basins can increase river baseflows[42] and maintain cool groundwater[43]. Utilizing underground storage shows promise for capturing reservoir spills produced, in part, from increasing minimum

reservoir storage[27] and requires little new infrastructure relative to dam construction.

The American West is in the midst of an ongoing megadrought[44], punctuated by wet periods[45]. Without a change in management to set environmental water demands as priority objectives, freshwater ecosystems downstream of large dams will be increasingly vulnerable to climate warming and related changes including declining snowpack, increased hydrologic volatility, warming stream temperatures, and shifting wet and dry seasonality[46,47]. Previous studies have shown that regulatory environmental flows are likely to be significantly affected by climate warming[40], and reservoirs will be relied upon to maintain environmental flows—especially during drought[22]. We advance environmental water management by demonstrating that allocating and managing reservoir storage for the environment is more efficient than mimicking downscaled natural flows. Droughts or other crises can be an impetus for improving water management to promote healthy communities and ecosystems[48]. For instance, Australia's Millennium Drought was a catalyst for improving the efficiency of environmental water entitlements and avoiding harm to ecosystems[15]. Allocating water and storage space to manage it in large, multipurpose reservoirs would provide a hedge against future drought and climate variability, and allow coordinated management of flow and habitat within and among watersheds[31].

## Methods

### Experimental reservoir overview

To examine environmental trade-offs with other water demands and understand temperature dynamics, we represented a large, multipurpose California reservoir using a simple priority-based water balance operations model coupled with a one-dimensional reservoir temperature model that stratifies vertically (Fig. 1). The experimental reservoir has a storage capacity of 5.55 billion cubic meters (Bm³), equal to Shasta Reservoir. This allowed us to use Shasta Reservoir inflow data (USBR's Shasta Dam station) for inflows, evaporation, outflows, and flood storage[27].

Releases from the model reservoir were represented in a simplified way with a temperature control device that has three openings. Minimum storage of 1.54 Bm³ was a constraint in some model runs. In effect, this expanded the "dead space" in the reservoir that could not be used to meet downstream demands and helped preserve cold water in the reservoir that could be accessed with a reservoir temperature control device. Minimum reservoir storage targets have been recommended for large dams. For instance, current operations of large reservoirs like Shasta Reservoir aim for minimum storage of 2.8 Bm³ by May 1[49] and about 1.54 Bm³ by October 1 to provide sufficient cold water to meet temperature objectives for salmonids[50].

### Environmental water demands

We developed three environmental water demand objectives based on the best available science: 1) environmental baseflows to account for minimum instream flows and water quality standards, 2) flow shaping to mimic aspects of a desired flow regime that support ecological processes and functions, and 3) optimal water temperatures to restore salmonid populations, which require colder water than needed for merely suitable temperatures. Environmental water demands have senior priority in our modeling and receive a percentage (10–40%) of reservoir inflow (discussed in Model Run section below).

**Environmental baseflows** are based on ecosystem water from the Delta water accounting in Gartrell et al.[21,22]., which attribute partially multipurpose reservoir releases into distinct "buckets" to fulfill water demands. Ecosystem water demands are primarily determined under the federal Clean Water Act and Endangered Species Act, and state law counterparts. We scaled ecosystem water

by the fraction of water that Shasta Reservoir contributes to the Delta. Environmental baseflows vary monthly and by water year type, reflecting regulatory requirements that supply more water to the environment in wetter years and less in drier years[21,22] (Supplementary Figure 2). Environmental baseflows average about 8% of reservoir inflow.

**Flow Shaping** provides seasonal volumes of water for a fall pulse, winter pulse, and spring recession. Water managers could shape magnitude, timing, duration, frequency, and rate of change through reservoir releases, an approach which is compatible with delivering functional flows or other prescribed environmental flows[7,8,30]. A spring recession flushes fine sediment and cues all runs of out-migrating juvenile Chinook Salmon, a fall pulse flushes fine sediment from spawning gravels and cues downstream movement of juvenile winter-run Chinook Salmon and upstream movement of returning fall-run Chinook Salmon, and a winter pulse cues downstream movement and inundates off-channel habitat utilized by diverse fish communities and all runs of salmon[31,51–53]. Flow shaping supplements environmental baseflows and averages about 14% of reservoir inflow volume. In our model, the total annual volume and the within-year distribution of flow-shaping demands remain constant each year. In practice, daily reservoir operations would likely alter flow shaping timing, magnitude, duration, frequency, and rate of change over the years to best meet downstream ecological objectives[54].

**The stream temperature objective** provides water temperatures optimal to enhance salmonid populations, with temperatures colder than 11.5 °C from June through December to improve winter-run egg and early fry survival, temperatures colder than 12.8 °C from December through April to improve pre-spawn survival for fall and late-fall Chinook Salmon runs that spawn in the mainstem Sacramento River, and temperatures less than 15 °C all year round to improve juvenile survival for all runs[55–57] (Supplementary Table 1).

### Other water demands

Water demands outside of the environmental demands include: 1) wildlife refuge water demands, 2) in-basin urban and agricultural uses, 3) system water for salinity maintenance, and 4) out-of-basin water exports. We assigned a timeseries of monthly demands that are defined as a function of the Sacramento River Index water year type (Supplementary Figure 3)[26,58]. For simplicity, hydropower generation and recreation were ignored.

The first three demands—refuge, in-basin urban and agricultural, and system water for salinity maintenance—share senior priority in the model, and demand varies depending on the time of year. These senior demands receive 60-90% of reservoir inflows (the remainder of the 10–40% allocated to environmental demands). The last—out-of-basin exports—is junior to the other demands. Despite our simplified accounting, water for some demands is multi-purpose—for example water to meet temperature standards could be reused to meet other demands[21].

**Wildlife refuge water demands** are separate from environmental demands because refuge water demands have water rights. Refuge demands in wet and above normal water years are equal to 684.6 million m³ (Mm³), while demands in below normal, dry, and critically dry water years are equal to 520.5 Mm³, as assigned by the federal Central Valley Project (CVP) Refuge Water Supply Program[59]. Monthly values were estimated using seasonal deliveries to wildlife refuges managed by the CVP[60]. Environmental water allocations do not augment refuge demands.

**In-basin urban and agricultural demands** provide water for cities and farms, which we combined for simplicity. Seasonal in-basin demands were modeled on CVP deliveries to the Sacramento Settlement Contractors and the Tehama-Colusa Canal[60]. These demands increase relative to other demands in drier years (Supplementary Figure 3).

**System water demands** are from the Delta water accounting study[21,22]. System water is Delta outflow necessary to meet salinity standards for in-Delta urban and agricultural uses and exports. While these flows also provide ecosystem benefits, ecosystem function is not the primary objective.

**Out-of-basin export demands** are modeled after observed pumping through the Tracy and Banks pumping plants located in the Delta[60]. Export demands are highest in wetter years, and significantly lower in critically dry years. These patterns reflect their junior water rights priority, which limits their access to water in dry years.

## Water balance model

We demonstrate the impact of environmental water assets, including dedicated storage for the environment, using deterministic water balance simulations designed to measure the ability of a reservoir to meet downstream demands, including environmental baseflow, temperature objectives, flow shaping, and other water demands with existing water allocations. The simplified water balance evaluates changes in reservoir storage subject to: (a) reservoir inflows, modeled after historical inflows into Shasta Reservoir, (b) monthly reservoir evaporation, modeled after historical reservoir evaporation from Shasta Reservoir, (c) reservoir releases to meet environmental and other demands, and (d) flood releases of any storage that encroached into the reservoir flood pool, as defined by US Army Corps of Engineers operating rules for Shasta Reservoir[61], such that:

$$S_{t+1} = S_t + I_t - E_t - RDD_t - RFC_t \qquad (1)$$

where $S$ is storage (af), $I$ is reservoir inflow (af/month), $E$ is reservoir surface evaporation (af/month), $RDD$ is released for downstream demands (af/month), $RFC$ is releases for flood control (af/month), and $t$ is the monthly timestep. We used measured historical inflows to Shasta Reservoir because historical inflows and unimpaired full natural flows into Shasta Reservoir were similar[27].

Average annual water demands for in-basin users, system water, and exports comprised the balance of non-flood control releases from Shasta Reservoir during the 26-year simulation period, such that:

$$D_{IB} + D_{SAL} + D_{EX} = \frac{1}{26}\sum_{t=WY1996}^{2021} R_t - ECO_t - WET_t - RFC_t \qquad (2)$$

where $D$ is total demand (af/year), $IB$ is in-basin demand (-), $SAL$ is system water demand (-), $EX$ is export demand (-), $R$ is total releases from Shasta Reservoir (af/month), $ECO$ is releases for environmental demands (af/month), $WET$ is releases for wetland refuge habitat (af/month), and $RFC$ is releases from Shasta Reservoir when the flood control pool is encroached upon (af/month).

On average, historical reservoir releases were split evenly among in-basin agricultural and urban demands, system water, and out-of-basin exports (e.g., $D_{IB} = D_{SAL} = D_{EX}$), although inter-annual and seasonal patterns reflected observed differences between the groups.

Flow shaping demands were added to environmental baseflow demands to create a two-tiered system of environmental water demands, where environmental baseflows were higher priority demands and flow shaping was considered lower priority. We did this to ensure that regulatory flows were maintained. When there was not enough water to meet all flow shaping objectives, water was allocated for the spring recession, then winter pulse, and finally the fall pulse flow.

To manage multipurpose operations within our modeled reservoir, each water demand group was designated a proportion of reservoir inflow and the same proportion of reservoir storage capacity (e.g., 10 % inflow and 10% storage capacity, 20% inflow, and 20% storage capacity, etc.). Within each capacity allocation, a water-demand-

specific water balance was conducted, such that:

$$S_{g,t+1} = S_{g,t} + k_g*(I_t - E_t) - RDD_{g,t} \qquad (3)$$

and

$$\sum_g k_g = 1.0 \qquad (4)$$

where $g$ is the water demand group (environmental, in-basin, system water, refuge, and exports) and $k$ is the inflow allocation to water demand group $g$.

Critically, storage for each water demand group ($S_g$) was not allowed to fall below 0, requiring water demands to experience delivery shortfalls when the volume of stored water was less than the monthly demand, such that:

$$RDD_{g,t} = \min(S_{g,t}, D_{g,t}) \qquad (5)$$

and

$$SF_{g,t} = D_{g,t} - RDD_{g,t} \qquad (6)$$

where $D$ is equal to the downstream demand of water demand group $g$ in timestep $t$ and $SF$ is equal to the delivery shortfall of water demand group $g$ in timestep $t$.

Stored water could be carried over for future use when capacity existed; however, carryover water was first to be spilled for flood control. Flood control releases, which are required when reservoir storage encroaches into the flood control pool, were divided among the storage accounts of each water demand. Responsibility for flood control releases was not assigned to all demands equally; instead, releases were assigned in proportion to the demand group's storage held in excess of their capacity allocation. This was represented as:

$$RFC_{g,t} = \max(S_{g,t} - c_g FC_t, 0.0) \qquad (7)$$

and

$$\sum_g c_g = 1.0 \qquad (8)$$

where $c$ is the capacity allocation assigned to demand group $g$ and $FC$ is the maximum flood control capacity of the hypothetical reservoir in timestep $t$.

When flood control conditions were triggered, deliveries to all demands were credited against the spilled water instead of reservoir storage accounts, and demand group storage was only impacted by their portion of the flood control release, such that:

$$S_{g,t+1} = S_{g,t} - RFC_{g,t} \qquad (9)$$

Reservoir storage and demand shortfalls were simulated for all five water demands for a range of pass-through, environmental storage space, and minimum storage alternatives. When results were analyzed for wet and dry periods, dry years include critically dry, dry, and below normal Sacramento River Index water year types, and wet years include above normal and wet year types. Reservoir storage volumes were subsequently linked to a one-dimensional reservoir temperature model, enabling simulations to evaluate how environmental storage could be used to manage trade-offs between downstream environmental demands and river temperature objectives.

## Water temperature model

Reservoir temperatures were estimated with Water Quality for Reservoir-River Systems (WQRRS), a mechanistic one-dimensional Fortran model developed originally by Chen and Orlob[62] and later

distributed by the US Army Corps of Engineers-Hydrologic Engineering Center[63]. Average monthly inflow, inflow stream temperature, and weather are the inputs. The model was run using a daily timestep, then averaged to a monthly timestep.

One-dimensional reservoir water quality models are appropriate for representing large reservoirs where water temperature changes most in the vertical direction based on atmospheric conditions and water density. We chose WQRRS because it runs quickly and has been widely used[64–67].

WQRRS is a finite difference model based on the principles of conservation of heat and mass. Heat and mass transfer vertically through advection and effective diffusion, and water was assumed perfectly mixed laterally and longitudinally. The reservoir was segmented into 90 vertical layers and each layer was 2 m deep, for a reservoir depth of 180 m. Water temperature was the only water quality constituent modeled, and was estimated using the heat budget method given the one-dimensional form of the advection-diffusion equation:

$$V\frac{\partial C}{\partial t} + \Delta x Q_x \frac{\partial C}{\partial x} = \Delta x A_x D_c \frac{\partial^2 C}{\partial x^2} + Q_i C_i - Q_o C \pm VS \qquad (10)$$

where $C$ is thermal energy (kcal), $V$ is volume (m³), $t$ is time (s), $x$ is vertical distance in the reservoir (m), $Q_x$ is advective flow (m³/s), $A_x$ is surface area (m³), $D_c$ is the effective diffusion coefficient (m³/s), $Q_i$ is lateral inflow (m³/s), $C_i$ is inflow thermal energy (kcal), $Q_o$ is lateral outflow (m³/s) and $S$ are sources and sinks (kcal/s).

Molecular and turbulent diffusion was based on temperature in WQRRS and convection was based on density gradient. Our hypothetical reservoir had one inflow at the upstream end of the reservoir, making the advection rate slower than if the inflow occurred near the dam. Inflows were instantaneously mixed within the reservoir layer of similar density[63]. Stratification was based on the relationship between density and water temperature.

Atmospheric conditions drove temperature exchange at the air-water interface and surface layer mixing. Inflow temperatures were from the Sacramento River at Delta (DLT) station (California Data Exchange Center). Air temperature, wind speed (m/s), and relative humidity (%) were from the Remote Automated Weather Station (RAWS) at Redding Airport for 2002–21 and Lincoln, California, prior to 2002. Atmospheric pressure was based on elevation and was constant at 29.15 Hg. Cloud cover (% of sky) was unavailable and was estimated to be uniform at 0.5%.

We represented a generalized temperature management infrastructure with a basic temperature control outlet. Outflows were modeled using the selective withdrawal allocation method developed by the US ACE Waterways Experiment Station to estimate the vertical limit of the withdrawal zone and vertical velocity distribution within that zone[63]. We modeled one withdrawal outlet with three opening ports and one spillway. The deepest withdrawal port was 25 m above the reservoir bed, the middle port was 65 m above the bed, and the upper port was 95 m above the bed. The spillway elevation was even with the surface of the dam when it was at capacity. In comparison, Shasta Reservoir has temperature control gates approximately 46 m, 61 m, 91 m, and 122 m above the reservoir bed, and the upper three gates have multiple shutters that can be opened to manage release temperatures.

## Model runs

Sixteen model runs were completed to understand performance on environmental objectives and to quantify trade-offs with other demands (Supplementary Table 2). Below we summarize model runs:

- **Pass-through of a percentage of inflow**. Four model runs represented pass-through of 10%, 20%, 30%, and 40% of inflows

with no minimum reservoir storage constraint. Four more runs represented 10–40% pass-through for the environment with 1.54 Bm³ minimum storage (Supplementary Table 2).
- **Percentage of inflow and percentage of storage capacity**. Four runs varied inflow allocations between 10 and 40% and allocated reservoir storage capacity by the same proportion, for example, pairing 10% inflow to 10% storage, 20% inflow to 20% storage, etc. (Supplementary Table 2). Four additional runs allocated the above inflow and storage capacity percentages with minimum storage constrained to 1.54 Bm³ to increase the likelihood of cold water in storage that could be accessed with a reservoir temperature control device.

## Data availability

The water balance input data used for this study are publicly available in the swfte_inputs folder of the SwftE GitHub repository: https://github.com/hbz5000/SwftE[68] The water balance data generated in this study have been deposited in the swfte_output folder of the same repository. The water temperature inputs and data generated by this study have been deposited in publicly available in the Storing Water for the Environment Hydroshare repository: https://www.hydroshare.org/resource/7dc98bc3f9bc498ea5a6ec4bbce3d60a/[69].

## Code availability

The water balance model, data, and code that support the findings of this study are publicly available in the following GitHub repository: https://github.com/hbz5000/SwftE[68]. The water temperature model, data, and code are publicly available in the following Hydroshare repository: https://www.hydroshare.org/resource/7dc98bc3f9bc498ea5a6ec4bbce3d60a/[69].

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

## Acknowledgements

This research was supported with funding from the S. D. Bechtel, Jr. Foundation and the funders of the Public Policy of California CalTrout Ecosystem Fellowship. AMS received salary support from UKRI [MR/V023578/1].

## Author contributions

Conceptualization: S.E.N., J.M., and B.G. Methods: S.E.N., H.Z., J.M., B.G., and A.M.S. Investigation: S.E.N. and H.Z. Visualization: S.E.N., H.Z., and G.S. Funding acquisition: J.M. and B.G. Supervision: S.E.N., J.M., and B.G. Writing—original draft: S.E.N. Writing—review & editing: S.E.N., H.Z., J.M., B.G., A.M.S., G.S., K.D., and B.T.

## Competing interests

The authors declare no competing interests.
