## [Peer Review File · Nature Communications]

Storing and Managing Water for the Environment is More Efficient than Mimicking Natural FlowsREVIEWER COMMENTS

Reviewer #1 (Remarks to the Author):

The authors analyze how a specific water-management system in California could be adjusted to more efficiently provide for multiple—often competing needs. The manuscript is well written and concise. The topic and conclusions are potentially of wide interest. However, I believe several improvements are necessary.

First, terminology needs more clarification and consistency of use. In the Introductory material, “environmental flows” and “environmental water” seem to be used interchangeably, which adds confusion. Line 49 refers to an “aquatic heat wave,” but no explanation is given. The term “functional flow objectives” is used on Line 98 and throughout the paper, but no explanation is provided as to what this entails. A brief explanation is needed for readers who are not intimately familiar with recent publications in that regard. The captions for Figures 3 and 4 are confusing. The lead sentence provides distinct categories of “dryness / wetness”, but those categories do not appear in the graphs.

I have several concerns about the Methods. Some may be addressed with more clarification, others may require rethinking of the analysis approach. First, there is no explanation given up front for the management scenarios described in Lines 75-76. A brief explanation would provide clarifying context. Are these realistic scenarios? My biggest concern is that the authors make inferences from Figures 3 and 4, but there are no estimates of uncertainty. This is puzzling given that each analysis was conducted (presumably) using information from a composite sample of years with observed data, each year having variation in precipitation and streamflow. Similarly, although the models may be deterministic (which I am unable to fully evaluate because I am not a hydrologic modeler), there is most certainly uncertainty associated with at least some of the parameters. This lack of uncertainty is especially concerning because the authors emphasize thresholds / breakpoints in their results. Declaring the presence of a threshold in the absence of uncertainty seems like a giant leap to me.

Reviewer #2 (Remarks to the Author):

Thank you for the opportunity to review this paper, it was an engaging exercise. Overall this paper presents a valuable and technically sound approach. The study demonstrates a simple, tractable approach for operating a reservoir by setting environmental water demand as a primary objective and explicit storage allocation rather than a constraint on operations, an important step towards integrated water management. It also expands on typical representation of environmental water targets by including minimum flows, seasonal flow components, and water temperature targets. The promising results suggest this could be a useful approach for identifying breakpoints in water demand trade-offs and increasing environmental water efficiency.

Specific comments:

- Key study aim: to demonstrate that allocating and managing reservoir storage for the environment is more efficient than mimicking downscaled natural flows.
 - o How novel is this contribution? More background on what has been done already in this space and how the current study builds on past research would help support this claim and contextualize the findings.
 - o This approach was demonstrated for one simplified large reservoir model - is that sufficiently representative to make this claim broadly? Could the authors justify this further and/or discuss in what settings this approach and results (e.g., specific tradeoff outcomes, breakpoints) are (not) expected to be applicable?
 - o The efficiency was demonstrated with respect to environmental flow volume but not other critical flow metrics, such as timing, rate of change, duration. Adding this component (flow timing in

particular) would enhance the study's novelty and help to more fully demonstrate the efficiency of the proposed approach. A single study of course cannot address all of the relevant pieces, but this seems like a large gap that could substantially affect reservoir allocation efficiency and at least warrants more discussion.

- Functional flows (FFs): A bit more description of FFs in the Introduction would help provide the reader with sufficient background to distinguish from environmental baseflows and interpret the performance results. After reading Methods and SI, it is still not clear if what was used in the study are really FFs, how these seasonal FF volumes were calculated, or where this information was obtained. Table S1 lists monthly flow volume thresholds and month ranges, but these do not align with the literature on FFs for California (see Yarnell et al. 2019; Patterson et al. 2020) or provide calculation methods. Please clarify.
- Model performance metrics: Including metrics beyond volumetric (% met), averaged by WYT, would be very helpful for evaluating performance across all the model objectives and runs considered. This could include frequency-based metrics (how often were targets met and in which months/years), or considering the worst performing or driest subset of years where performance may otherwise be obscured by averaging.
- Figures 3/4 – This may just be personal preference, but I would have liked to see wet and dry years on the same or adjacent plots - even if a bit busier - for direct comparison (maybe dashed or different shading lines?)
- Supplemental figures – please clarify all parts of key (e.g. W, AN, etc) in figure captions

Citations noted:

Yarnell, S., Stein, E.D., Webb, J.A., Grantham, T., Lusardi, R., Zimmerman, J., Peek, R., Lane, B., Howard, J. and Sandoval, S. 2019. "A functional flows approach to selecting ecologically relevant flow metrics for environmental flow applications," *River Research and Applications*. DOI: 10.1002/rra.3575

Patterson, N., Lane, B., Sandoval-Solis, S., Pasternack, G.B., Yarnell, S., & Qiu, Y. 2020. "A hydrologic feature detection algorithm to quantify seasonal components of flow regimes," *Journal of Hydrology*. <https://doi.org/10.1016/j.jhydrol.2020.124787>

Reviewer #3 (Remarks to the Author):

Overall, I think this paper provides an important perspective on implementation of environmental flows. The topic is important and timely and the modeling/analysis are sound. However, I have several strong concerns about how environmental flow needs are defined and quantified, which has implications for the paper's results and conclusions. Environmental flow needs aren't straightforward to define. In fact, a key reason why we have so many declining freshwater species is that regulatory flows in rivers are either never defined or are negotiated to be much lower than necessary to support ecological function, but these insufficient regulatory flows are then used to determine whether ecological needs are met and how much water is available for other uses. In short, most current approaches to determining environmental flows are inadequate and that point needs to be addressed in any paper that assesses trade-offs in flows available for beneficial uses.

In the introduction, the authors describe a few common variations of how to define environmental flows: a) functional flows, b) designer flows, and c) pass-through flows. In the rest of the paper, the authors state that they are comparing two different types of flow approaches; a) pass-through flows and b) a proportion of reservoir inflow plus a proportion of reservoir storage allocated to the environment. The authors then compare results of these two approaches to metrics of attaining environmental flow objectives: a) regulatory flow requirements, b) functional flows, c) temperature targets.

My main criticism is that the concept of functional flows is used incorrectly in the paper, unless I

completely misunderstood how it was applied by the authors. Functional flows as an environmental flows approach have been explicitly defined by a series of publications and tools on the California Environmental Flows Framework (CEFF), and quantitative flow targets can be downloaded for rivers in California at <https://flowline.codefornature.org>. The concept of functional flows is meant to be an alternative to inadequate methods of determine flow needs for the environment. It took reading the entire paper, and through the entire methods including the supplemental material, to determine that the authors didn't model functional flows at all as defined by CEFF, rather they modeled designer flows that are meant to improve very specific aspects of a small part of a hydrograph, as described in Supplemental Table 1. In addition, the three studies meant to represent functional flows represents only a small sample of the literature available that might be used to design flows in the Sacramento River. For example, recent papers have shown that flow magnitude strongly affects survival for Chinook salmon. Michel et al. (2019) showed that over a third of all variability in the smolt-to-adult ratio could be explained by flow during outmigration for all three Central Valley Chinook populations. Cordoleani et al. (2018) showed that wetter years with higher flows strongly benefited survival of spring run juveniles migrating out from Butte Creek. Notch et al. (2020) documented dramatically higher survival for spring run outmigrants in a year with higher flow. Michel et al. (2021) estimated that optimal spring flows for outmigrants were ~10,000 cfs at Wilkins Slough. Just to list a few. None of these papers were included in defining flow needs, and all of these papers suggest an increasing benefit to salmon with increasing flows, not a cap at environmental benefit at 30% of inflows as suggested in the paper.

This is important because designer flows have always been the default method to define regulatory flows in California, since they focus on very small portions of the hydrograph meant to achieve one type of benefit (usually based on habitat) for one life-history stage for one listed species, and ultimately lead to less water allocated to the environment. In contrast, functional flows are intended to provide flow conditions that support river processes that broadly support freshwater biodiversity, and by definition include flow prescriptions for all parts of the hydrograph, including variability within and across year types. A functional flows approach would start with prescribed flows using existing tools, supplemented and possibly refined by additional information about specific flows needs for listed species. But regardless of additional flow needs in the literature, flow prescriptions would be required for the full year.

Lines 261-267 show the shortcomings of this approach. Although the authors said the management scenarios are largely hypothetical, they are also explicitly based on operations at Shasta Dam, environmental flow regulations for the Sacramento Valley and San Francisco Bay-Delta watershed, and papers on ecological flow needs from this same geographic area. The authors explicitly state that there would be no increasing environmental benefit of flow allocations above 30%, which is much lower than any evidence or analyses of flow needs in the Sacramento River watershed. For example, the 10,000 cfs necessary to provide optional outmigration flows at Wilkins Slough cited by Michel et al (2021) represents a flow value much higher than 30% of reservoir inflow. An analysis performed by the State Water Board found that 75% of flow would be needed to achieve salmon objectives in the Sacramento River and Delta (California State Water Resources Control Board 2010).

Specifically, I recommend either 1) changing the name of "functional flows" as applied in the paper to "designer flows", or 2) create a set of functional flows using the CEFF approach that applies to the full-year hydrograph. Alternatively, the authors could explain that the flows that they are labeling as "functional flows" in their analysis do not represent a full functional flows approach to defining environmental flow needs, but can be used as an example of how inflows could be stored and shaped to achieve specific flow outcomes.

Putting the issue of defining environmental flow needs aside, I do think the analysis in the paper is interesting and important, and does a good job at highlighting the benefits of storing water and releasing in a pattern different from unimpaired inflows to address temperature requirements of species and functional flows downstream of dams.

RESPONSE TO REVIEWERS

Thank you to each reviewer for your thoughtful comments. We have addressed each comment to improve and clarify our manuscript. Below we describe changes we made in blue, with reviewer comments in black. Line numbers refer to the clean version of the manuscript.

REVIEWER COMMENTS

Reviewer #1 (Remarks to the Author):

The authors analyze how a specific water-management system in California could be adjusted to more efficiently provide for multiple—often competing needs. The manuscript is well written and concise. The topic and conclusions are potentially of wide interest. However, I believe several improvements are necessary. First, terminology needs more clarification and consistency of use. In the Introductory material, “environmental flows” and “environmental water” seem to be used interchangeably, which adds confusion.

We revised so that environmental water refers to water for any environmental objective (e.g., flows, temperature management, etc.). We retaining environmental flows only when describing literature that evaluates or recommends streamflow prescriptions for environmental protection.

Line 49 refers to an “aquatic heat wave,” but no explanation is given.

We removed this text from the manuscript since it is not the focus of our study.

The term “functional flow objectives” is used on Line 98 and throughout the paper, but no explanation is provided as to what this entails. A brief explanation is needed for readers who are not intimately familiar with recent publications in that regard.

We changed our term ‘functional flow objectives’ to ‘flow shaping objectives’ and define it as “*water for a fall pulse, winter pulse, and spring recession [that] can be shaped by water managers to mimic aspects of flow regimes that support ecological function.*” (line 82). We removed mention of functional flows in response to comments from all reviewers.

The captions for Figures 3 and 4 are confusing. The lead sentence provides distinct categories of “dryness / wetness”, but those categories do not appear in the graphs.

We added “*Dry years include critically dry, dry, and below normal Sacramento River Index water year types, and wet years include above normal and wet year types*” To Figure 3, Figure 5 (previously Figure 4) and in methods (line 508-510).

I have several concerns about the Methods. Some may be addressed with more clarification, others may require rethinking of the analysis approach. First, there is no explanation given up front for the management scenarios described in Lines 75-76. A brief explanation would provide clarifying context. Are these realistic scenarios?

These are realistic scenarios. We moved examples of these approaches from the discussion to the introduction: “*These alternatives are exemplified by new flow objectives for California’s San Joaquin River. The California State Water Resources Control Board adopted amendments that require water users to pass-through or release an average of 40% of February–June unimpaired flows on the Lower San Joaquin River and its tributaries if water users fail to negotiate ‘voluntary agreements’ to reduce water use that is approved by the Board¹⁷. Further upstream, the San Joaquin River Restoration Program has a Restoration Administrator who manages a percentage of unimpaired inflow, which can be stored and released to provide ecosystem benefits¹⁸.*” (lines 68-74)

My biggest concern is that the authors make inferences from Figures 3 and 4, but there are no estimates of uncertainty. This is puzzling given that each analysis was conducted (presumably) using information from a composite sample of years with observed data, each year having variation in precipitation and streamflow.

We added inter-annual uncertainty/variability in Figures 3 and 5 (previously Figure 4). We originally omitted variability ranges to keep figures simple. These figures now visualize annual variability to environmental and other water demands. Hydrologic variability incorporates variability from precipitation, air temperature, and other climate variables.

Similarly, although the models may be deterministic (which I am unable to fully evaluate because I am not a hydrologic modeler), there is most certainly uncertainty associated with at least some of the parameters. This lack of uncertainty is especially concerning because the authors emphasize thresholds / breakpoints in their results. Declaring the presence of a threshold in the absence of uncertainty seems like a giant leap to me.

We added that our models are deterministic (line 435) to clarify our approach.

We tried to walk the line between showing yearly uncertainty in our results to ground our findings, while avoiding refocusing this paper as a sensitivity analysis of hydroclimatic uncertainty, which is less novel (see for example, Zhen et al. 2013; Mishra 2009; Benke et al. 2008). Since we added interannual variability in Figures 3 & 5, and show monthly and yearly storage, deliveries, and operations during drought conditions in Figure 6, we opted to keep Figure 7 unchanged. When inter-annual uncertainty is added to Figure 7, it becomes hard to understand.

Our finding that storing water for the environment shifts tradeoff curves toward a more efficient solution/compromise and necessitates smaller environmental water allocations has not been described in the literature and could improve multi-objective water management. We better highlighted this novelty in the text and added quantitative estimates of shortages for competing water uses (see lines 268-283). We also reorganized our text so this section is now followed by discussion of the simplicity of our model and our recommendation to use existing, more sophisticated models to better elucidate potential tradeoffs and breakpoints in real systems.

Benke, K.K., Lowell, K.E. and Hamilton, A.J., 2008. Parameter uncertainty, sensitivity analysis and prediction error in a water-balance hydrological model. *Mathematical and Computer Modelling*, 47(11-12), pp.1134-1149.

Mishra, S., 2009. Uncertainty and sensitivity analysis techniques for hydrologic modeling. *Journal of hydroinformatics*, 11(3-4), pp.282-296.

Zhan, C.S., Song, X.M., Xia, J. and Tong, C., 2013. An efficient integrated approach for global sensitivity analysis of hydrological model parameters. *Environmental Modelling & Software*, 41, pp.39-52.

Reviewer #2 (Remarks to the Author):

Thank you for the opportunity to review this paper, it was an engaging exercise. Overall this paper presents a valuable and technically sound approach. The study demonstrates a simple, tractable approach for operating a reservoir by setting environmental water demand as a primary objective and explicit storage allocation rather than a constraint on operations, an important step towards integrated water management. It also expands on typical representation of environmental water targets by including minimum flows, seasonal flow components, and water temperature targets. The

promising results suggest this could be a useful approach for identifying breakpoints in water demand trade-offs and increasing environmental water efficiency.

Specific comments:

- Key study aim: to demonstrate that allocating and managing reservoir storage for the environment is more efficient than mimicking downscaled natural flows.
- How novel is this contribution? More background on what has been done already in this space and how the current study builds on past research would help support this claim and contextualize the findings.

We revised the intro to highlight novelty and the contribution of our study. We omitted the section that describes different types of environmental flows and instead focus on understanding the efficiency of environmental water (where more previous research focuses on the effectiveness of environmental flows) (Paragraph 2 in the Introduction). We then can move quickly into how our work builds on past research (Paragraph 3 in the Introduction).

We show that less water is required to meet environmental water objectives if storage capacity is available/allocated to best manage environmental water. This advances environmental water management by focusing on assets besides water for environmental protection.

- This approach was demonstrated for one simplified large reservoir model - is that sufficiently representative to make this claim broadly? Could the authors justify this further and/or discuss in what settings this approach and results (e.g., specific tradeoff outcomes, breakpoints) are (not) expected to be applicable?

We specify that *“our approach applies to large, multi-purpose reservoirs, with highly variable seasonal and interannual inflows, temperature stratification during summer, minimum operational levels (dead pool), and seasonal flood storage requirements”* (lines 98-100).

We also reorganized the tradeoff section to follow our breakpoint analysis with a discussion of limitations *“Our model is simple, intended as a proof-of-concept to understand and compare how portions of inflow and storage capacity allocated to the environment could benefit ecosystem objectives and impact other water demands. Our approach complements studies that prescribe environmental or functional flows^{7,8,29}. It is not intended to be a guide for setting specific standards or determining the adequacy of environmental flows to support species and ecosystem function. Sophisticated water management and water temperature models exist for California’s water system³⁴⁻³⁷ and most large river basins^{38,39}. Those models could be applied to scrutinize and elucidate the potential benefits, tradeoffs, breakpoints, and impacts of inflows-plus-storage space allocation in real systems.”* (lines 297-304)

Our results suggest that allocating and managing reservoir storage for the environment is promising. In light of our findings—and when possible-- it makes sense to focus some effort on managing storage for environmental objectives.

- The efficiency was demonstrated with respect to environmental flow volume but not other critical flow metrics, such as timing, rate of change, duration. Adding this component (flow timing in particular) would enhance the study's novelty and help to more fully demonstrate the efficiency of the proposed approach. A single study of course cannot address all of the relevant pieces, but this seems like a large gap that could substantially affect reservoir allocation efficiency and at least warrants more discussion.

Our approach set aside seasonal blocks of “*water for a fall pulse, winter pulse, and spring recession (Sup. Table 1), [which] can be shaped by water managers to mimic aspects of flow regimes that support ecological function.*” (line 82-84). In other words, the timing, rate of change, duration... can be managed as long as the volume of water in each season is not exceeded.

In the methods, we further clarified that “*Water managers could shape magnitude, timing, duration, frequency, and rate of change through reservoir releases, an approach which is compatible to deliver functional flows or other prescribed environmental flows^{7,8,29}*” (lines 387-389).

- **Functional flows (FFs):** A bit more description of FFs in the Introduction would help provide the reader with sufficient background to distinguish from environmental baseflows and interpret the performance results. After reading Methods and SI, it is still not clear if what was used in the study are really FFs, how these seasonal FF volumes were calculated, or where this information was obtained. Table S1 lists monthly flow volume thresholds and month ranges, but these do not align with the literature on FFs for California (see Yarnell et al. 2019; Patterson et al. 2020) or provide calculation methods. Please clarify.

We used the literature to estimate the required water to meet some river functions, but we did not actually develop functional flows. For the reason, we changed our term ‘functional flow objectives’ to ‘flow shaping’ (e.g., line 82). We omitted the paragraph describing functional flows and other types of environmental flows to better focus the introduction.

- **Model performance metrics:** Including metrics beyond volumetric (% met), averaged by WYT, would be very helpful for evaluating performance across all the model objectives and runs considered. This could include frequency-based metrics (how often were targets met and in which months/years), or considering the worst performing or driest subset of years where performance may otherwise be obscured by averaging.

We added environmental baseflow shortages (m^3/yr), flow shaping shortages (m^3/yr), and summer (July – September) reservoir releases ($^{\circ}C$) as additional metrics to evaluate performance. This added figure 4, with metric values incorporated into the text (e.g., lines 118-120, 185-187).

The ‘Example Environmental Water Storage Performance in 2019-2021’ considers the driest subset of years when performance would otherwise be obscured by averaging. We expanded this figure by adding a pass-through scenario to compare with the proportion of inflow and storage capacity scenario. We also added percentage and volumetric water shortages into this section.

- **Figures 3/4 –** This may just be personal preference, but I would have liked to see wet and dry years on the same or adjacent plots - even if a bit busier - for direct comparison (maybe dashed or different shading lines?)

We placed wet and dry years in adjacent plots in Figures 3 & 5 for direct comparison. We also added interannual variability as shading around the means to visualize uncertainty.

- **Supplemental figures –** please clarify all parts of key (e.g. W, AN, etc) in figure captions
We defined year type acronyms in the supplemental figure captions.

Reviewer #3 (Remarks to the Author):

Overall, I think this paper provides an important perspective on implementation of environmental flows. The topic is important and timely and the modeling/analysis are sound. However, I have several strong concerns about how environmental flow needs are defined and quantified, which has implications for the paper's results and conclusions. Environmental flow needs aren't straightforward to define. In fact, a key reason why we have so many declining freshwater species is that regulatory flows in rivers are either never defined or are negotiated to be much lower than necessary to support ecological function, but these insufficient regulatory flows are then used to determine whether ecological needs are met and how much water is available for other uses. In short, most current approaches to determining environmental flows are inadequate and that point needs to be addressed in any paper that assesses trade-offs in flows available for beneficial uses.

We incorporated these thoughts about the limitations of regulatory requirements in the 2nd paragraph *"When implemented, water is typically withheld from appropriation to farms and cities to comply with water quality, flow, and endangered species regulatory requirements or negotiated compromises. This makes those flow requirements a constraint on water operations, rather than a priority objective in multipurpose water management"*.¹¹ (line 38-41).

We address this concern in our approach by separating baseflows *"to account for existing minimum instream flows and water quality standards"* (line 81-82), then builds on that by providing seasonal blocks of *"water for a fall pulse, winter pulse, and spring recession [that] can be shaped by water managers to mimic aspects of flow regimes that support ecological function"* (line 82-84). We renamed the latter a 'flow shaping objective', changed from 'functional flows' in our previous version, to better match our method with correct terminology.

In the introduction, the authors describe a few common variations of how to define environmental flows: a) functional flows, b) designer flows, and c) pass-through flows. In the rest of the paper, the authors state that they are comparing two different types of flow approaches; a) pass-through flows and b) a proportion of reservoir inflow plus a proportion of reservoir storage allocated to the environment. The authors then compare results of these two approaches to metrics of attaining environmental flow objectives: a) regulatory flow requirements, b) functional flows, c) temperature targets. My main criticism is that the concept of functional flows is used incorrectly in the paper, unless I completely misunderstood how it was applied by the authors. Functional flows as an environmental flows approach have been explicitly defined by a series of publications and tools on the California Environmental Flows Framework (CEFF), and quantitative flow targets can be downloaded for rivers in California at <https://flowline.codefornature.org>. The concept of functional flows is meant to be an alternative to inadequate methods of determine flow needs for the environment. It took reading the entire paper, and through the entire methods including the supplemental material, to determine that the authors didn't model functional flows at all as defined by CEFF, rather they modeled designer flows that are meant to improve very specific aspects of a small part of a hydrograph, as described in Supplemental Table 1.

We see the reviewer's point. We changed our term 'functional flow objectives' to 'flow shaping objectives', defined as *"water for a fall pulse, winter pulse, and spring recession [that] can be shaped by water managers to mimic aspects of flow regimes that support ecological function"* (line 82-84). Our paper focuses on passing water through reservoirs or using reservoir storage to meet environmental water objectives. We did not develop functional flows as explicitly defined. Rather, our approach would be complimentary if/when functional flows have been defined (line 299-300). In addition, the three studies meant to represent functional flows represents only a small sample of the literature available that might be used to design flows in the Sacramento River. For example, recent papers have shown that flow magnitude strongly affects survival for Chinook salmon. Michel et

al. (2019) showed that over a third of all variability in the smolt-to-adult ratio could be explained by flow during outmigration for all three Central Valley Chinook populations. Cordoleani et al. (2018) showed that wetter years with higher flows strongly benefited survival of spring run juveniles migrating out from Butte Creek. Notch et al. (2020) documented dramatically higher survival for spring run outmigrants in a year with higher flow. Michel et al. (2021) estimated that optimal spring flows for outmigrants were ~10,000 cfs at Wilkins Slough. Just to list a few. None of these papers were included in defining flow needs, and all of these papers suggest an increasing benefit to salmon with increasing flows, not a cap at environmental benefit at 30% of inflows as suggested in the paper. We added references to Michael et al. 2021, and Michel 2019, Cordoleani et al. 2018, and Michel et al. 2023. Since we shifted description of our environmental objective from functional flows to flow shaping, so we made no attempt to revise for a thorough summary of all functional flow literature.

This is important because designer flows have always been the default method to define regulatory flows in California, since they focus on very small portions of the hydrograph meant to achieve one type of benefit (usually based on habitat) for one life-history stage for one listed species, and ultimately lead to less water allocated to the environment. In contrast, functional flows are intended to provide flow conditions that support river processes that broadly support freshwater biodiversity, and by definition include flow prescriptions for all parts of the hydrograph, including variability within and across year types. A functional flows approach would start with prescribed flows using existing tools, supplemented and possibly refined by additional information about specific flows needs for listed species. But regardless of additional flow needs in the literature, flow prescriptions would be required for the full year.

Since we have clarified our that our flow shaping objective uses seasonal blocks of water to mimic distinct aspects of desired flow regimes to support ecological processes and functions, we believe we have addressed the substance of this comment. We added that carryover storage could be used to provide higher flows in some years (line 257-239). Finally, we added that our method is compatible with functional flow approaches (line 387-389).

Lines 261-267 show the shortcomings of this approach. Although the authors said the management scenarios are largely hypothetical, they are also explicitly based on operations at Shasta Dam, environmental flow regulations for the Sacramento Valley and San Francisco Bay-Delta watershed, and papers on ecological flow needs from this same geographic area. The authors explicitly state that there would be no increasing environmental benefit of flow allocations above 30%, which is much lower than any evidence or analyses of flow needs in the Sacramento River watershed. For example, the 10,000 cfs necessary to provide optional outmigration flows at Wilkins Slough cited by Michel et al (2021) represents a flow value much higher than 30% of reservoir inflow. An analysis performed by the State Water Board found that 75% of flow would be needed to achieve salmon objectives in the Sacramento River and Delta (California State Water Resources Control Board 2010).

This comment highlights the why our findings are exciting. Using 30% of inflow and 30% of storage capacity, environmental water can be carried over and flows shaped so that releases can sometimes be much higher than 30% (provided environmental demands are lower in other years/seasons). As a result, water is used *more efficiently* than dedicating 75% of flow all of the time, when flows of that magnitude are needed only sometimes. The pass-through flows proposed by the California Water Resources Control Board are an example of using water *effectively*, but *inefficiently*. To better highlight this finding, we added: “allocating inflows with storage capacity to manage water allows environmental water to be used more efficiently than pass-through. These alternatives shift tradeoff curves leftward in Figure 7, toward a more optimal region with greater total benefit. For example, with 20% pass-through, 48% of flow objectives are unmet. The shortage drops to 21-26% when

storage is used to manage environmental water, depending on minimum reservoir storage for cold-pool.” (line 274-279)

Specifically, I recommend either 1) changing the name of “functional flows” as applied in the paper to “designer flows”, or 2) create a set of functional flows using the CEFF approach that applies to the full-year hydrograph. Alternatively, the authors could explain that the flows that they are labeling as “functional flows” in their analysis do not represent a full functional flows approach to defining environmental flow needs, but can be used as an example of how inflows could be stored and shaped to achieve specific flow outcomes.

We changed the name from functional flows to flow shaping to capture that we set aside blocks of water for a fall pulse, winter pulse, and spring recession (line 96-102). Those volumes of water could be managed to alter the frequency, timing, magnitude, duration, and rate of change of reservoir releases for downstream ecosystems.

Putting the issue of defining environmental flow needs aside, I do think the analysis in the paper is interesting and important, and does a good job at highlighting the benefits of storing water and releasing in a pattern different from unimpaired inflows to address temperature requirements of species and functional flows downstream of dams.

Thank you!

REVIEWERS' COMMENTS

Reviewer #1 (Remarks to the Author):

The authors have adequately addressed the concerns I raised in my first review.

Reviewer #1 (Remarks on code availability):

None

Reviewer #2 (Remarks to the Author):

Thank you for addressing my comments, the manuscript has been much improved.

Note that Figure 3 still says 'functional flows' rather than 'flow shaping'

Response to Reviewers (NCOMMS-23-44553A)

Reviewer 1 had no actionable comments or revisions. Reviewer 2 had one comment. Our response is in blue below.

Reviewer #1 (Remarks to the Author):

The authors have adequately addressed the concerns I raised in my first review.

Reviewer #1 (Remarks on code availability):

None

Reviewer #2 (Remarks to the Author):

Thank you for addressing my comments, the manuscript has been much improved.

Note that Figure 3 still says 'functional flows' rather than 'flow shaping'

We have revised the legend in Figure 3 to read 'flow shaping' rather than 'functional flows.'